



# Data quality control and calibration for mini-radiosonde system "Storm Tracker" in Taiwan

Hung-Chi Kuo[1], Ting-Shuo Yo[1], Hungjui Yu[2], Shih-Hao Su[3], Ching-Hwang Liu[3], Po-Hsiung Lin[1]

[1]Department of Atmospheric Sciences, National Taiwan University, Taipei, Taiwan
[2]Department of Atmospheric Science, Colorado State University, Fort Collins, Colorado, USA
[3]Department of Atmospheric Sciences, Chinese Culture University, Taipei, Taiwan

*Correspondence to*: Ting-Shuo Yo (tsyo@ntu.edu.tw)

**Abstract.** This study introduced and evaluated the calibration schemes of a newly developed upper-air radiosonde instrument, "Storm Tracker" (ST), with data collected in field observations during 2016–2022. The ST is a radiosonde instrument developed and tested in 2016 (Hwang et al., 2020). In a series of field campaigns in the Taiwan area, more than one thousand co-launches of ST and Vaisala RS41-SGP (VS) are conducted. Using the VS measurements as the reference, we developed data correction methods and examined the characteristics of the ST sounding. The corrected ST soundings have 1-K temperature and 7% relative humidity root mean square difference to the VS soundings. These error differences can be reduced to 0.66-K and 4.61% below the 700-hPa height. The GPS estimated ST wind error difference is about 0.05 ms-1. The results suggested that the ST can perform similarly to the reference sounding and has reached the level required for environmental sampling and scientific research. The geostrophic adjustment dynamics indicate that the spatial temperature variation in the free atmosphere may not be large. However, the lower atmosphere may have large wind, temperature, and moisture variations. Due to the relatively low cost and accuracy after correction, ST can complement regular upper-air observations for high spatial and temporal resolution.

## 1 Introduction

With over a hundred years of history, upper-air radiosondes are one of the crucial meteorological instruments and the most reliable one to gather atmospheric data at various altitudes. The measured pressure, temperature, and relative humidity (so-called "PTU") data aids in weather forecasting, climate research, and the study of atmospheric dynamics. However, upper-air radiosondes are subject to certain biases due to instrument calibration, ascent rates, and environmental conditions. Collins (2001) distinguished the radiosonde observational errors into three types: random, rough, and systematic. According to Collins (2001), random error is caused by small-scale turbulence or unsystematic observational errors, and it is impossible to correct. The rough error can be introduced from observational protocol, computational error for data processing, or communication-related error. A properly defined operational procedure and automatic quality control process can minimize such errors. The third type of error, systematic error, is caused by insufficiencies in measurement devices or data processing procedures and persists in all observational data. This type of error can be detected and calibrated with statistical methods.



Nowadays, commercial radiosondes are often tested and corrected regarding these biases. However, they are typically characterized by their higher weight and cost, which limit the deployment of scientific field campaigns. The independently developed mini-radiosonde system – the "Storm Tracker (referred to as the ST, Figure 1b)" was developed and first tested in 2016 (Hwang et al., 2020). The ST was then put into intensive field observation operations for the first time during the Taipei

Summer Storm Experiment (TASSE) in 2018–2020. The main goal of the field campaign is to investigate the thermal characteristics of the boundary layer in the Taipei Basin and local wind field variations to improve the forecasting ability of afternoon convection in the metropolitan area. Three advantages of using the ST for atmospheric field research were learned. First, the weight with the battery of only 20g for ST helps with helium/hydrogen usage. Second, the commercial sensors, chips, and signal transmission components in the ST significantly reduce the cost and provide flexibility for multiple deployments

and high spatial and temporal resolution observations. Lastly, the ST is easy to set up and can be quickly deployed or even mobile, which provides adaptability for different research needs and broadens the possibility for field campaign design.

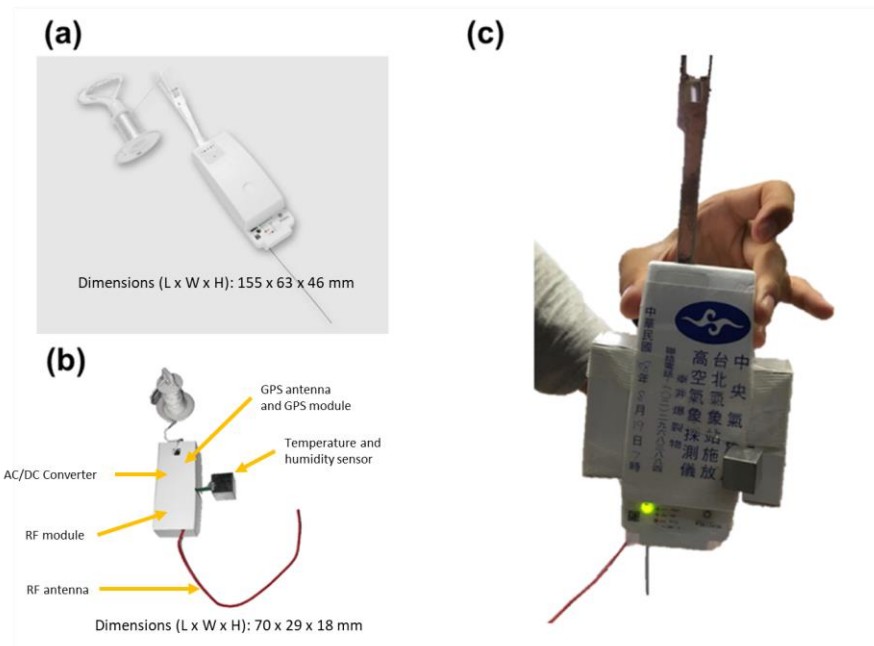

**Figure 1. (a) The Vaisala RS41-SGP radiosonde (weighted 84g, body dimension: 155 x 63 x 46 mm), (b) the storm tracker mini-**
**radiosonde (weighted 20 g with battery, body dimension: 70 x 29 x 18 mm), and (c) an example of the co-launched soundings via the TASSE experiment. More ST hardware details are described in Hwang et al. (2020).**

During TASSE, the ST measurements showed an overall warm and dry bias in the troposphere compared to the VS. Figure 2 shows an example of such a bias pattern. These biases result from a well-recognized issue, as Vömel et al. (2007) suggested, that solar radiation can induce warm and dry bias for radiosonde measurements. Similar daytime warm and dry biases have

been reported in previous field experiments around the world that used relatively mature radiosonde systems (e.g., Wang et





al., 2002; Ciesielski et al., 2009; Yu et al., 2015). Earlier studies indicated that radiosonde temperature biases are primarily contributed by radiative effects, with a minor proportion caused by the sensor response lag of the changing of temperatures as the radiosonde rises (e.g., McMillin et al., 1992; Sun et al., 2013).

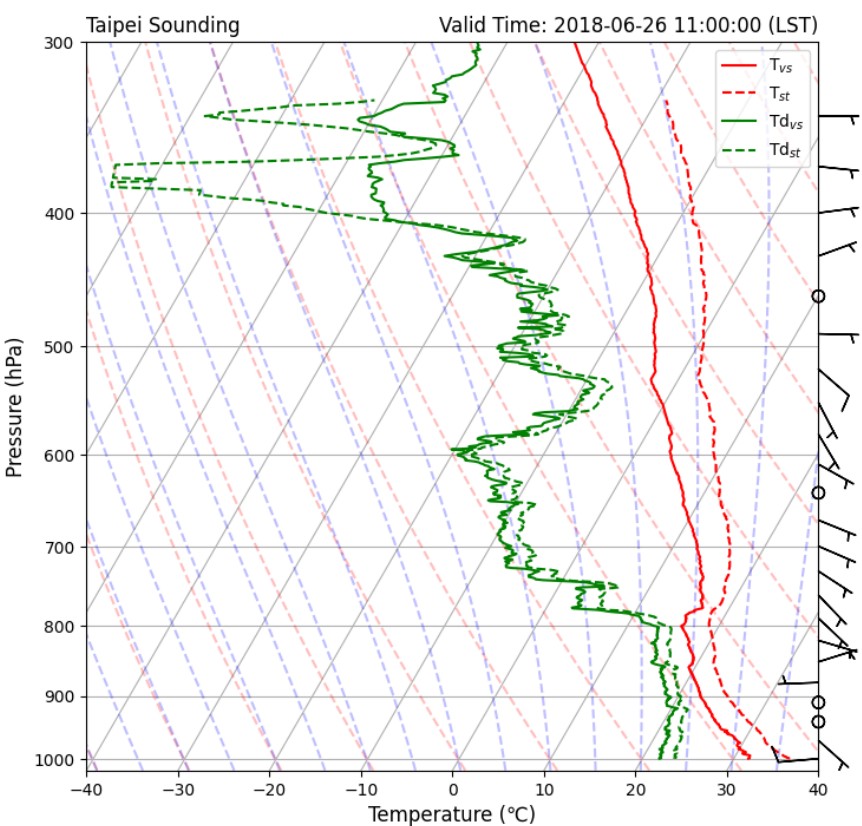


**Figure 2. The sounding of 2018-06-26 11:00 LST by VS (solid lines) and ST (dashed lines). The ST profile showed warm and dry bias near the surface.**

The daytime temperature bias induced by solar heating was identified with various radiosonde systems (e.g., Luers, 1989, 1997; Luers and Eskridge, 1998; Sun et al., 2013). Their findings resulted in special surface coating over temperature sensors in most commercial radiosondes. Even though environmental parameters can still affect the observed temperature, all factors influencing radiative or sensible heat flux around the sensor, such as the sensor surface temperature, solar angle, cloud fraction, and ventilation velocity, can cause the sensor temperature bias (e.g., McMillin et al., 1992; Luers and Eskridge, 1995; Mattioli
et al., 2007). Luers and Eskridge (1998) evaluated the impact of the environmental parameters on the radiosonde in detail.





Their results suggested that the temperature bias is most sensitive to solar angle, while the cloud cover has a slight impact. Also, the ventilation effect may cause bias when the sensor is in the balloon wake zone.

In addition to temperature bias, the humidity bias has been discussed in many studies (e.g., Vömel et al., 2007; Yoneyama et al., 2008; Nuret et al., 2008). Vömel et al. (2007) found that the solar-heating-induced dry bias increased with altitude in the

troposphere, which means the humidity bias also depended on the temperature. This resulted in the relative humidity (RH) measured in the low-temperature environment being less accurate (Miloshevich et al., 2001). Miloshevich et al. (2004) also pointed out that the response delay in humidity sensors could cause measurement errors at low temperatures. The influence of these biases could be huge. For example, in the Tropical Ocean Global Atmosphere Coupled Ocean-Atmosphere Response Experiment (TOGA COARE,1992-1993), scientists have reported the observational error induced an unrealistically dry

boundary layer and caused an underestimate of convective available potential energy (CAPE) (Miller et al., 1999; Lucas and Zipser, 2000). Although the primary observation targets of ST are the lower troposphere environmental conditions, we still noticed significant warm and dry deviations in the near-surface boundary layer in TASSE (Figure 2).

Many studies have attempted to remedy the systematic error in radiosonde data with statistical methods. Lesht and Richardson (2002) mentioned that Vaisala accounts for the sensitivity of the RH sensor to temperature by using a high-order polynomial

function with empirical coefficients. Yoneyama et al. (2008) applied a polynomial fitting function of pressure for the relative difference of RH and used the solar zenith angle as a factor for bias corrections. Other studies leveraged the thermodynamic equation and provided the temperature correction table with empirical correction factors (Wang et al., 2013; Dzambo et al., 2016).

In past field campaigns, scientists have also developed the statistical model of humidity correction based on probability

matching. For example, Ciesielski et al. (2009) used the cumulative distribution function (CDF) matching method to correct the humidity bias for nearby soundings. The advantage of the CDF-based calibration method is that the calibration procedure is fast and straightforward. Building the correction table requires sufficient data to represent the statistical characteristics and questionable data can be adjusted to match the same distribution. The basic concept of the CDF matching calibration method is assuming the ambient atmospheric conditions are similar for all observation sites. In most field campaigns, the spatial

distribution of upper-air radiosonde sites mostly satisfied such requirements, and hence, this method can efficiently adjust the data bias for most atmospheric conditions. However, such assumptions limit the generalizability of the CDF calibration models. Thus, the CDF models may not be directly applied to the data collected from different weather conditions, seasons, or climate regions with smaller sample sizes.

Although wind speed and direction are crucial information in radiosonde observation, we found from the co-launched data that

the GPS-estimated ST wind differs from that of VS in insignificant magnitude. In this study, we focused on the calibration process of systematic error for ST temperature and moisture observations using the co-launch VS data. We use the co-launch data collected across several field campaigns in Taiwan to develop calibration methods for ST. Here, we proposed and evaluated two different calibration approaches. First, we followed the widely used CDF-matching approach and proposed a two-step CDF-based calibration scheme. Secondly, we incorporated the CDF-matching approach with modeling multivariate



distributions, the central concept of machine learning, to introduce a novel correction method based on the generalized linear model (GLM). While the CDF approach discretized continuous variables, e.g., pressure and temperature, into bins to establish look-up tables, the machine-learning approach modeled a high-dimensional joint probability distribution with the same variables in their original forms. The latter approach allowed us to compress complicated look-up tables into a unified mathematical representation. Hence, we can adjust the models more easily for better performance, robustness, and generalizability.

The co-launched radiosonde data, algorithms of bias correction methods, and data calibration processing flow are described in Section 2. In section 3, the results of the ST calibration are summarized and compared to the benchmark. Finally, the feature importance analysis and other calibration issues are discussed in Section 4, and conclusions are drawn in Section 5.

## 2. Data and Preprocessing

### 2.1 Data Collection

In the previous years since 2018, we have co-launched the ST with the Central Weather Bureau (CWB) operational Vaisala RS41-SGP radiosonde (Figure 1c). The co-launch was conducted during field campaigns in the Taiwan area, including the Taipei Summer Storm Experiment (TASSE), the Yilan Experiment of Severe Rainfall (YESR2020), the Taiwan-Area Heavy Rain Observation and Prediction Experiment (TAHOPE), the Northern Coast Observation, Verification, and Investigation of Dynamics (NoCOVID21), and the Mountain Cloud Climatology (MCC) project, We collected 1,029 co-launches of ST and VS from these field campaigns during 2018–2022. These co-launches provided more than 1,000,000 comparable independent observations of wind, pressure, temperature, and humidity (PTU) data. The numbers of co-launches of each campaign are summarized in Table 1

Table 1. The summary of the field experiments conducting ST-VS co-launches.

| Experiment | Time | Location | Total Numbers of ST-VS Co-launch |
|---|---|---|---|
| Taipei Summer Storm Experiment (**TASSE**) | 2018-2020 | Taipei (Banqiao) | 478 |
| Yilan Experiment of Severe Rainfall (**YESR2020**) | 2020. Nov | Yilan, Suaou, Luodong, Dafu | 46 |
| Taiwan-Area Heavy rain Observation and Prediction Experiment (**TAHOPE**) | 2019-2022 | Taipei (Banqiao), Pengjiayu | 382 |
| Northern Coast Observation, Verification, and Investigation of Dynamics (**NoCOVID21**) | 2021. May-Jun | Taipei (Banqiao) | 49 |
| Mountain Cloud Climatology (**MCC**) | 2022. Oct-Nov | Suaou | 23 |
| Other | | Tainan, Xinwu | 51 |





In 2018 and 2019, based on the scientific goals of TASSE, we established a standardized procedure for the co-launches, and the observations were primarily conducted in the daytime. Once the observational procedure matured, we performed the day

and night co-launches evenly in 2020, 2021, and 2022. (Table 2). Eventually, we collected 625 daytime cases and 404 nighttime cases. Also, the pilot experiments were conducted in the summer, and in the latter field experiments, we performed the co-launches in other months. Though there were more cases in July and August, we still conducted at least 21 co-launches in May. As for the location, most co-launches were conducted at the Taipei weather station, while about 150 cases were in other cities in Taiwan.


**Table 2. The summary of the 1,029 co-launches.**

| Month | 2018 | | 2019 | | 2020 | | 2021 | | 2022 | | Total | |
|---|---|---|---|---|---|---|---|---|---|---|---|---|
| | Day | Night | Day | Night | Day | Night | Day | Night | Day | Night | Day | Night |
| 1 | 0 | 0 | 0 | 0 | 21 | 24 | 0 | 0 | 25 | 26 | 46 | 50 |
| 2 | 0 | 0 | 0 | 0 | 29 | 29 | 0 | 0 | 0 | 0 | 29 | 29 |
| 3 | 0 | 0 | 0 | 0 | 27 | 31 | 0 | 0 | 0 | 0 | 27 | 31 |
| 4 | 0 | 0 | 0 | 0 | 30 | 30 | 15 | 13 | 0 | 0 | 45 | 43 |
| 5 | 0 | 0 | 0 | 0 | 6 | 5 | 6 | 4 | 0 | 0 | 12 | 9 |
| 6 | 14 | 0 | 20 | 0 | 0 | 0 | 30 | 32 | 0 | 0 | 64 | 32 |
| 7 | 14 | 0 | 60 | 12 | 0 | 0 | 22 | 23 | 0 | 0 | 96 | 35 |
| 8 | 41 | 0 | 85 | 0 | 0 | 0 | 23 | 22 | 0 | 0 | 149 | 22 |
| 9 | 0 | 0 | 0 | 0 | 0 | 0 | 25 | 26 | 0 | 0 | 25 | 26 |
| 10 | 0 | 0 | 0 | 0 | 0 | 0 | 29 | 28 | 7 | 3 | 36 | 31 |
| 11 | 0 | 0 | 0 | 0 | 20 | 17 | 40 | 41 | 6 | 7 | 66 | 65 |
| 12 | 0 | 0 | 0 | 0 | 0 | 0 | 30 | 31 | 0 | 0 | 30 | 31 |
| Total | 69 | 0 | 165 | 12 | 133 | 136 | 220 | 220 | 38 | 36 | 625 | 404 |

## 2.2 Pre-processing of the co-launch data

The ST is with the wind estimated from GPS. We analyzed the difference in wind variables with the paired data of VS and ST.

The mean deviation in zonal and meridional wind components, u and v, are 0.04 and 0.03 ms-1, respectively. The difference may come from the time lag of GPS signals between two sensors, which is small enough to ignore. We emphasize the correction of temperature and humidity calibration in this paper.

The co-launch's primary purpose is to understand ST's performance further and develop a data correction scheme to approximate the VS's observations. The raw data collected often contains inconsistencies, inaccuracies, and outliers that can



significantly distort analytical results and impede the accuracy of predictive modeling. Therefore, we need a proper procedure
        to process the raw data.

        In the work of Ciesielski et al. (2012), the authors suggested four stages for developing research-quality radiosonde data (their
        figure 1). The first level requires a single unified data format. The second stage uses automated tools to remove unreliable data
        based on prior knowledge of quality control (QC) checks. Then, data biases are detected and corrected in the third level based

on analysis or statistical methods. Finally, the fourth level dataset aims to be user-friendly, usually in uniform vertical
        resolution with QC flags.

        Following the framework proposed by Ciesielski et al. (2012), our data correction method is applied in the third stage. Hence,
        we need a pre-processing scheme to derive a level 2 dataset from the raw co-launch data.

        Figure 3 illustrates the preprocessing used in this study. In the first stage. First, we paired each ST and VS observation by

nominal observation time and stored them in the same plain-text format, L1_ST and L1_VS. Then, in the second stage, we
        corrected known errors for both sensors, including missing values and outliers. After this stage, we derived the level 2 dataset,
        L2_ST and L2_VS. Finally, given the fact that both ST and VS radiosondes are attached during co-launch (as Figure 1c), we
        used "time after launch" (every second) in both profiles to pair the values of two sensors, and resulted in L2_ST-VS.

        Based on the prior studies of ST (Hwang et al., 2020), we performed a "ground check" procedure to correct the pressure values

of ST. This procedure adjusts the P_ST by a constant bias dP_0, which is the difference between the surface pressure of the
        standard instrument and the sensor of ST. Furthermore, we filtered out profiles with inconsistent timestamps and records less
        than 250. Finally, we derived a dataset of 663 merged profiles and 1,219,710 paired entries (up to every second) for further
        analysis.

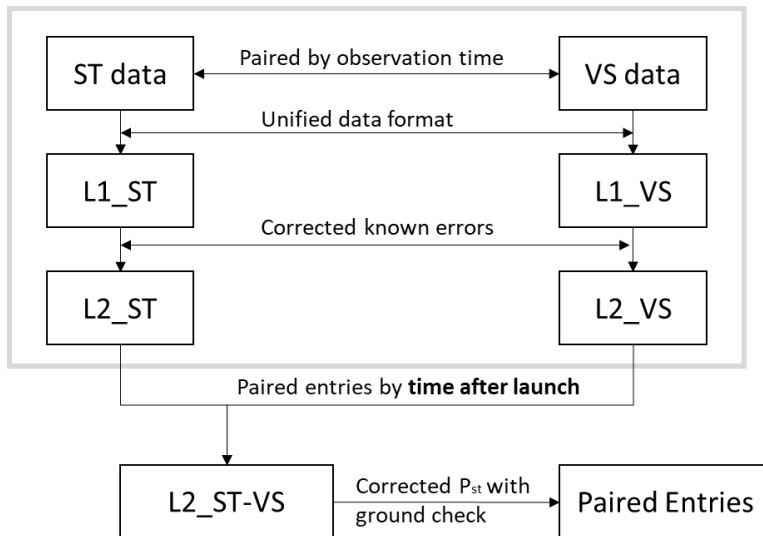






**Figure 3. The preprocessing for ST and VS data from raw to level 2.**

## 3. Data Correction Methods

To develop a data correction scheme for ST, we first investigated the conventional CDF-based probability matching method (Ciesielski et al., 2009). Then, we extended this approach with direct modeling of multivariate distributions, which is the central concept of modern machine learning. We implemented the scheme with the basic generalized linear model (GLM) and compared the differences between the two approaches.

Before diving into the specific correction methods, we define the notations and symbols used in this study. While ST and VS represent the storm tracker and the Vaisala RS41-SGP radiosonde device, respectively, they are used as subscripts to denote the sensor of measurements. For example, PST means the pressure measured by ST, and TVS is the temperature recorded by VS. The Δ(delta) symbol is used to denote the difference of the same variable between two sensors. Finally, the ' (prime) represents the corrected measure.

### 3.1 CDF-based Probability Matching

CDF-based Probability matching, also known as histogram matching or quantile mapping, is a statistical technique used to adjust the distribution of a dataset (e.g., a forecast distribution) to match that of another dataset (e.g., an observed distribution). The primary objective of this method is not to directly correct individual data points but to ensure that the overall statistical properties, such as the frequency of occurrence of specific values, match between the two datasets. In radiosonde observation, CDF-based probability matching is commonly used as a quality control tool to ensure data quality consistency for field campaigns (Nuret et al., 2008; Ciesielski et al., 2009).

Based on the paired entries collected in co-launches, the two-step correction scheme starts with correcting temperature (ΔT) based on the ground-checked pressure ($P'_{ST}$) and the measured temperature ($T_{ST}$). Then, the adjusted temperature ($T'_{ST}$) is used together with the measured relative humidity ($RH_{ST}$) to estimate the correction (ΔRH).

We first discretize the pressure and temperature variables in temperature correction into bins. Pressure is divided into 50 hPa intervals from 975–1025 hPa to 175–225 hPa, denoted by their centers, 1000 hPa to 200 hPa. Temperature is rounded to integers and forms 1-degree intervals from -80 to 40 degrees Celsius. For each pressure bin, we calculate the cumulative distribution function of temperature measured by ST and VS. Based on the assumption that two sensors have the same CDF within this specific range, we derived the correction values, ΔT, as a function of measured temperature, $T_{ST}$. Figure 4 demonstrates the CDF-based temperature correction of the pressure bin 475–525 hPa. The upper panel shows the CDF of $T_{VS}$ and $T_{ST}$, and the lower panel illustrates the correction (ΔT) as a function of the observed temperature ($T_{ST}$). We grouped the co-launches into daytime and night-time and performed the above procedure for each pressure bin. The results are shown in Figure 5, the complete temperature correction table used in this study.





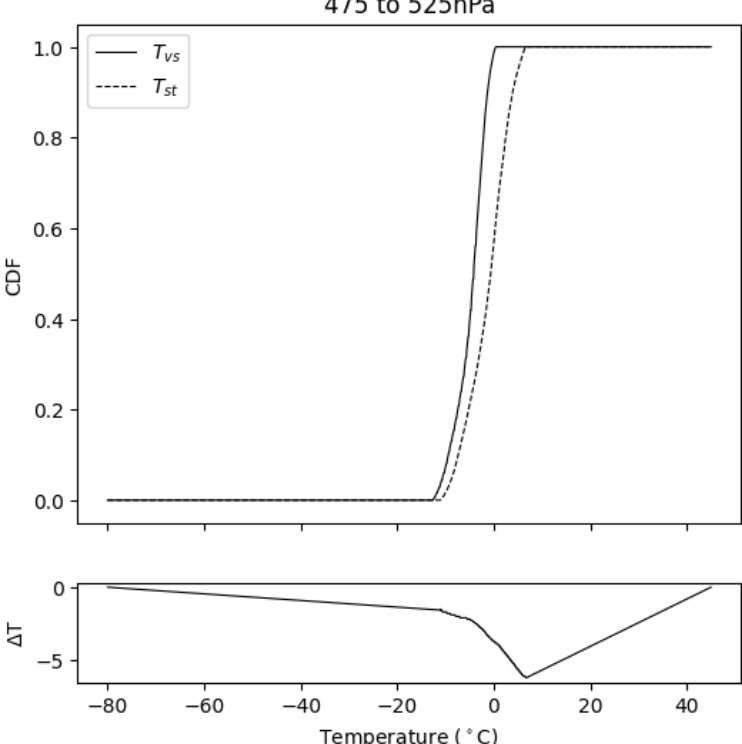


**Figure 4. The CDF-based temperature correction of the pressure bin 475 ~ 525 hPa. The upper panel shows the CDF of the temperature of two sensors, and the lower panel shows their difference as a function of temperature.**

As shown in Figure 5, the temperature sensor of ST consistently shows warm bias in all pressure bins, and the bias is stronger
at high altitudes. The night-time warm bias exhibits similar patterns to the daytime but with a lower quantity.

The correction of relative humidity (RH) is derived in the same way as the temperature, except for the independent variables, which are the corrected temperature ($T'_{ST}$) and the measured relative humidity ($RH_{ST}$). The corrected temperature is discretized into 10-degree intervals from -65 to 35 degrees Celsius. The relative humidity values are then rounded to integers and form 1% intervals from 0 to 100. Like the temperature correction procedure, the correction value is derived based on the CDF
probability matching as a function of RH within each temperature bin. Figure 6 illustrates the complete RH correction table used in this study. Figure 6 indicates that the ST shows dry-bias in lower altitudes and wet-bias in higher altitudes. ST is generally dryer during the daytime.

Using the correction tables shown in Figures 5 and 6, the temperature and relative humidity measured by ST are corrected and evaluated. Mathematically, this procedure can be expressed as:

$\Delta T = T_{VS} - T_{ST} = f(P'_{ST}, T_{ST}, Day)$ (1)

$\Delta RH = RH_{VS} - RH_{ST} = f(T'_{ST}, RH_{ST}, Day)$ (2)



, where Day is a binary variable represents the daytime or night-time, and f is the CDF-based probability matching.

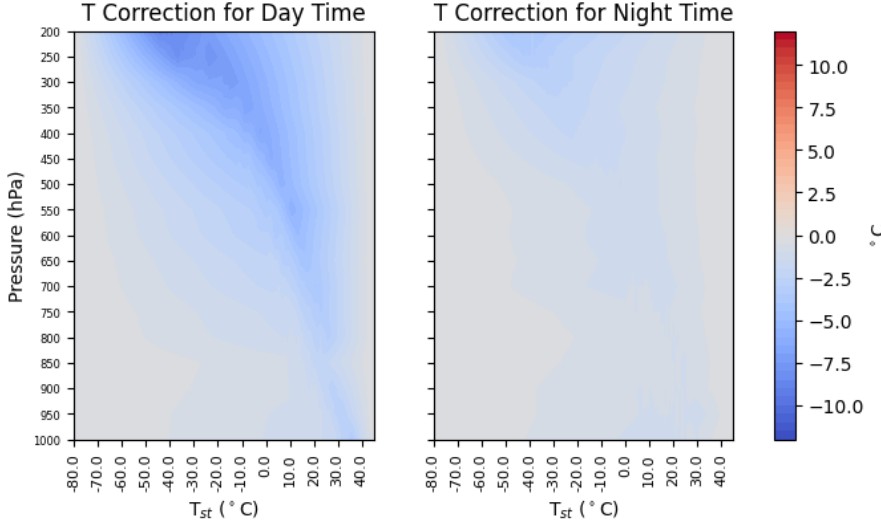


**Figure 5.** The CDF-based temperature correction tables for daytime (00z-12z, left panel) and night-time (12z-00z, right panel).

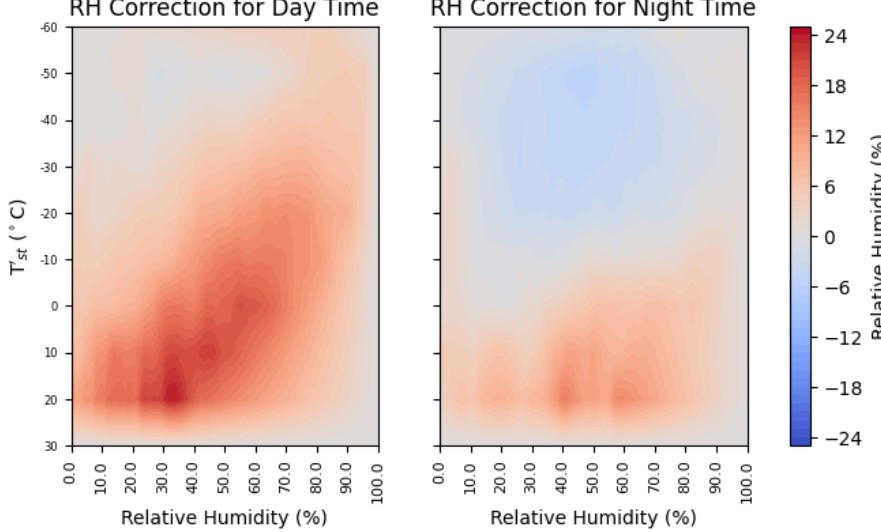

**Figure 6.** The CDF-based RH correction tables for daytime (00z-12z, left panel) and night-time (12z-00z, right panel).





## 3.2 Generalized Linear Model

Despite the robustness and ease of implementation of CDF-based probability matching, the discretization steps and the form of the look-up table limit its application. For example, the discretization of pressure and temperature is empirical. Though the resulting CDFs and correction tables look reasonable, it is hard to justify that this is the only way to split a continuous variable into bins. In other words, by focusing on matching the overall distribution, probability matching may overlook or alter some of the finer-scale details in the dataset. Furthermore, the look-up table makes adding extra independent variables more complicated. For example, we used daytime and night-time tables to simplify the influence of solar radiation so that we could use two tables for each correction. Another example is when we consider adding the effect of pressure in the correction of RH. In that case, we need to establish tree-dimensional bins and justify whether the cut-off points are adequately selected. Therefore, we want to introduce the modeling of the multivariate probability distribution to our correction scheme.

In essence, modeling the joint probability distributions of multiple variables is fundamental in machine learning for capturing relationships and dependencies among numerous predictors. It forms the backbone for various algorithms and techniques to predict, generate, and understand multi-dimensional data. In equations (1) and (2), the mapping function, f, can be seen as a model of the joint probability distribution of the independent variables. While the CDF-based probability matching algorithm models this distribution by discretizing the independent variables, it can be replaced by different algorithms that keep the predictors in their continuous form.

The Generalized Linear Model (GLM, Nelder and Wedderburn, 1972) is a versatile statistical framework used for modeling the relationship between a dependent variable (response) and one or more independent variables (predictors) in a wide range of applications. GLMs extend the concept of linear regression to handle a broader array of data types and distributions. They are particularly valuable for offering interpretable coefficients for understanding the impact of predictors on the response. GLMs have become a fundamental tool in statistics and data analysis due to their flexibility and applicability across various fields. In this study, we used GLMs in three different settings: the same scheme as CDF-based probability matching (as specified in equations (1) and (2)), using the same set of predictors for T and RH corrections, and relacing daytime with Julian-day and hour-of-day.

To develop the GLM-based corrections, we simply used the paired entry dataset and the least squared algorithm to fit linear regression models for the response variables ($\Delta T$ and $\Delta RH$) and the predictors ($P'_{ST}$, $T_{ST}$, $RH_{ST}$, and Day). This study used the Python algorithm implementation from scikit-learn (Pedregosa et al., 2011). The resulting regression equations are used to correct the storm tracker data.

In our second GLM configuration, we used the same independent variables, i.e., $P'_{ST}$, $T_{ST}$, $RH_{ST}$, and Day, to predict the corrections of temperature ($\Delta T$) and relative humidity ($\Delta RH$). The resulting models can be mathematically denoted as:

$$\Delta T = f(P'_{ST}, T_{ST}, RH_{ST}, Day) \tag{3}$$

$$\Delta RH = f(P'_{ST}, T_{ST}, RH_{ST}, Day) \tag{4}$$





As mentioned, many studies have suggested that solar radiation could be the leading cause of the warm bias in the radiosonde data. This is why we established correction tables for daytime and night-time separately. To simplify the correction process and limit the number of tables created, the solar radiation is represented by the binary variable, Day. However, with GLMs, we can easily use continuous variables in their original form. Hence, we used the "Julian day from the summer solstice" (Jday) and the "hour-of-day from noon" (Hour) to replace the Day variable. The resulting models are:

$$\Delta T = f(P'_{ST}, T_{ST}, RH_{ST}, Jday, Hour) \tag{5}$$

$$\Delta RH = f(P'_{ST}, T_{ST}, RH_{ST}, Jday, Hour) \tag{6}$$

These three settings are noted as GLM1, GLM2, and GLM3 in the later text.

## 4. Experiment Results

Figure 7 illustrates the patterns and deviations between ST and VS at various pressure levels. The panels (a), (b), and (c) demonstrate the temperature of VS and ST, and the differences between the two sensors. The relative humidity is shown in panels (d), (e), and (f). As shown in Figure 7, the ST exhibits warm and dry biases in general, and the biases increase as the altitude rises.



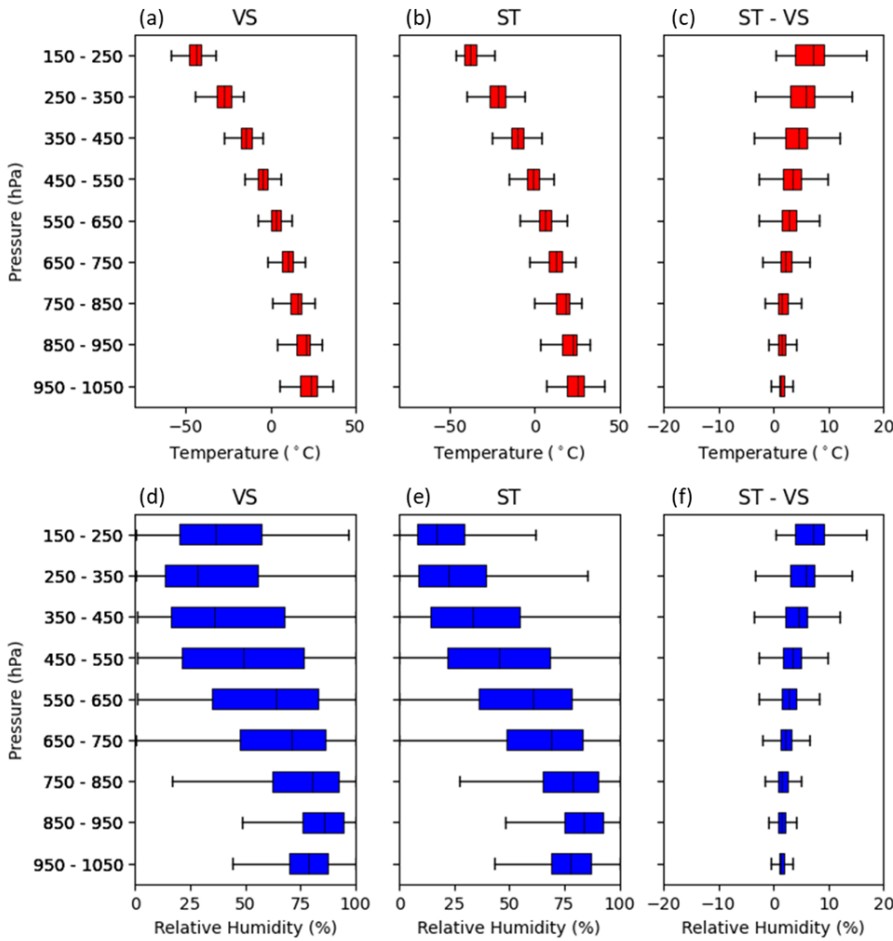

**Figure 7. The boxplot of temperatures (upper) and RH (lower) of ST (left), VS (center), and their difference (right).**

We applied the four correction methods described in the previous section, i.e., CDF, GLM1, GLM2, and GLM3, to the 663 sounding profiles. Using the VS as the reference observations, we calculated the root-mean-squared errors (RMSEs) as the evaluation metrics. We did not use the correlation coefficients for evaluation because two sensors have correlation coefficients higher than 0.99, even without corrections. The reason for this lies in the co-launching strategy, which ensures that both instruments endure the same environmental conditions. The means and standard deviations of RMSEs for all correction methods are shown in Table 3 and Figure 8. As shown in Figure 8, we can see a significant bias reduction for all correction methods. We performed t-tests on the raw and corrected values, and the improvement of all four methods is statistically significant (for p-values little than 10e-29). We also compared the CDF and GLM, and the results show that CDF correction is slightly better than GLMs for both temperature and relative humidity. The difference between CDF and GLMs is significant in the t-test, though the significant level is much lower than their bias reduction.



We also conducted t-tests on different GLM settings. The GLM1 and GLM2 did not show significant differences in temperature
and relative humidity correction results. However, the GLM3 showed significant improvement compared to GLM1 and GLM2.

280 This suggested that solar radiation parameters can influence the correction more than a simple day/night indicator.

Table 3 and Figure 8 also show the evaluations for all records below 500- and 700-hPa heights. As shown in the results, ST
can proximate the VS measurements with a temperature error of less than 1 degree Kelvin and a relative humidity error of less
than 10%. Suppose we focus on the observations below 700 hPa. In that case, the average error can be as low as 0.66-degree
Kelvin for temperature and 4.61% for relative humidity, comparable to the uncertainties of VS temperature and relative

285 humidity measurements (Vaisala, 2017).

**Table 3. The RMSE of ST and VS with different correction methods for temperature and RH.**

| Variable | Correction Method | mean RMSE | | | stdev of RMSE | | |
|---|---|---|---|---|---|---|---|
| | | full | 500hPa | 700hPa | full | 500hPa | 700hPa |
| Temperature | Uncorrected | 2.9969 | 2.0753 | 1.6446 | 1.8399 | 1.2291 | 0.8894 |
| | CDF | 0.8778 | 0.7568 | 0.6560 | 0.5579 | 0.4166 | 0.3367 |
| | GLM1 | 1.2714 | 1.1126 | 1.0732 | 0.6612 | 0.4549 | 0.3682 |
| | GLM2 | 1.2745 | 1.1128 | 1.0533 | 0.6625 | 0.4633 | 0.3693 |
| | GLM3 | 1.1991 | 1.0105 | 0.9483 | 0.6284 | 0.4566 | 0.3579 |
| Relative Humidity | Uncorrected | 8.5265 | 6.0721 | 4.9336 | 3.8236 | 2.9284 | 2.3624 |
| | CDF | 6.8946 | 5.4707 | 4.6098 | 2.8107 | 2.7488 | 2.4442 |
| | GLM1 | 7.4604 | 5.8673 | 4.9267 | 2.9158 | 2.6489 | 2.3084 |
| | GLM2 | 7.4152 | 5.7997 | 4.8478 | 2.7785 | 2.4307 | 2.0590 |
| | GLM3 | 7.2683 | 5.6355 | 4.7043 | 2.6668 | 2.3372 | 1.9878 |





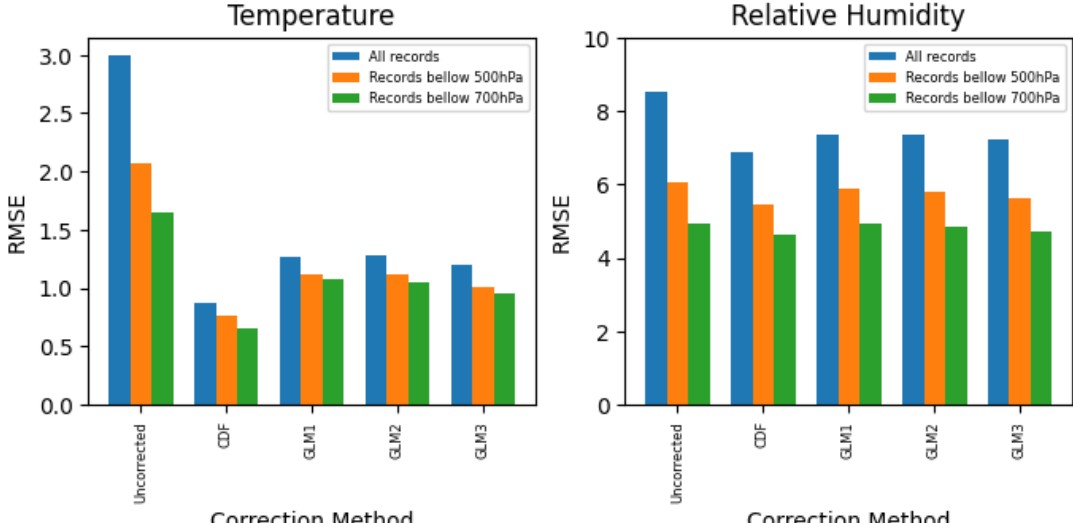

**Figure 8. The mean RMSE of ST and VS with different correction methods for temperature (left) and RH (right). For each correction method, the mean RMSE is derived with all available records (blue), records below 500 hPa (orange), and records below 700hPa (green).**

## 5. Discussion

### 5.1 General performance of ST

Figure 9 illustrates the paired entries of VS and ST before and after corrections. As described in the previous section, the ST exhibits correlation coefficients higher than 0.99 for temperature and RH even before any correction. Hence, the effect of corrections is represented by the narrower diagonals in the right panels in Figure 9.





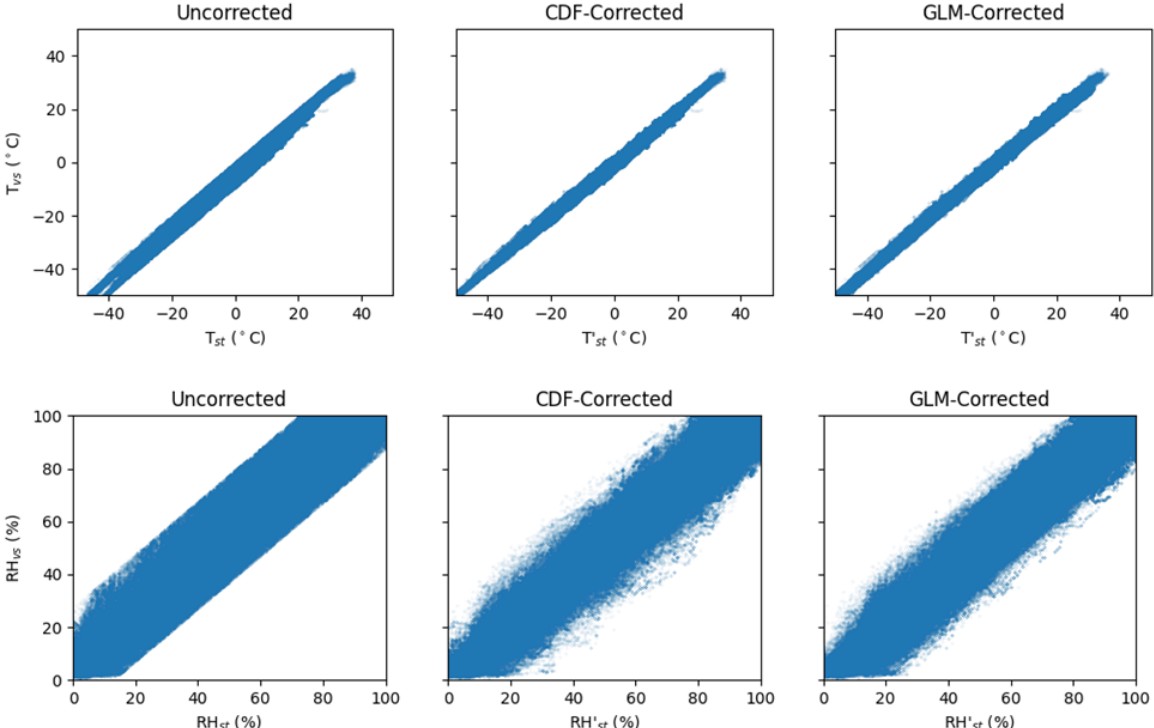

**Figure 9. The scatter plots of temperature (upper) and RH (lower) before and after correction.**

Though the statistical tests showed the significance of the correction results, they are not easily perceived. Hence, we selected a few sounding profiles to demonstrate the effectiveness of our correction methods. Figure 10 shows the T and RH profile of the sounding launched at 2021-08-03 12Z. This sounding was selected because of the overall low RH bias before and after correction. In Figure 10, the corrected temperature is adequately aligned to the reference ($T_{VS}$), and the corrected relative humidity (RH) is entirely satisfactory, particularly below 350hPa, covering most tropospheric levels with water vapor and clouds. Consistent findings are prevalent within our dataset, indicating that the adjusted ST measurements are reliable across various observational scenarios.



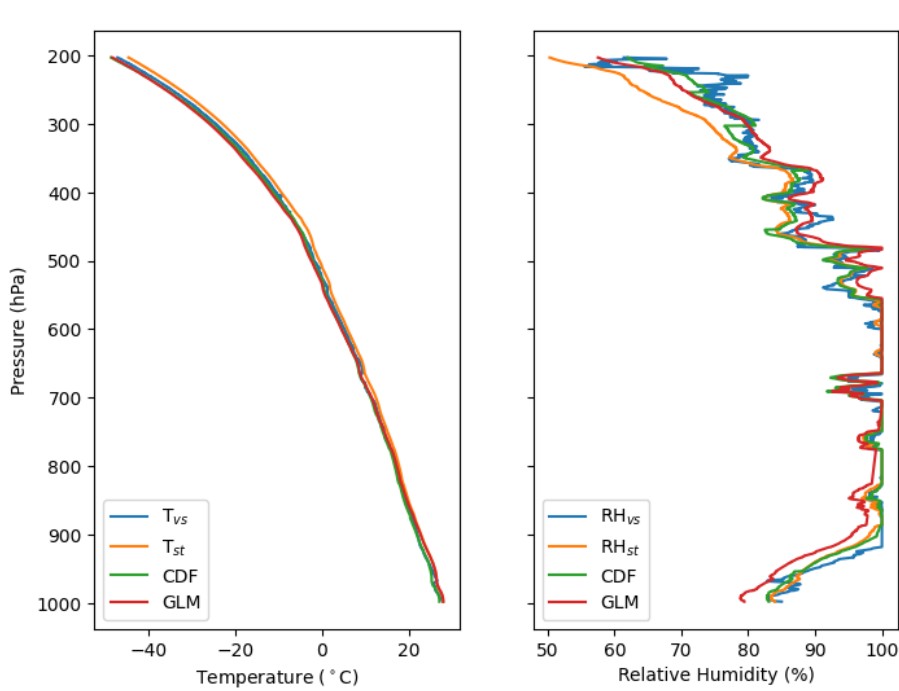


**Figure 10. The temperature (left) and RH (right) of the co-launch on 2021-0803 12Z. The reference (VS) is illustrated in blue, the ST in range, CDF-corrected in green, and GLM-corrected in red.**

However, the corrected results may perform less when encountering extreme wet cases. Figure 11 is the sounding profile on 2018-08-27 06Z when the reference RH of VS is about 90% from ~850 to ~350-hPa heights. As shown in Figure 11, the temperature correction still works properly. However, the RH measured by ST shows a dry bias of magnitude of 20% from ~850 to ~350-hPa heights while the patterns stay similar. The RH correction mechanisms adjust the RH toward the reference, but the deviations are still significant. Note that this observation occurred during a severe rainfall event caused by the

convergence of the tropical depression and the southwest monsoon from August 23 to August 30, 2018. All fifteen co-launches conducted in this event exhibited high bias in RH, ranging from 10% to 24%, and five showed bias greater than 10% even after correction. This particularly biased case has RMSE ranked 99.93% in our dataset.



2018082706

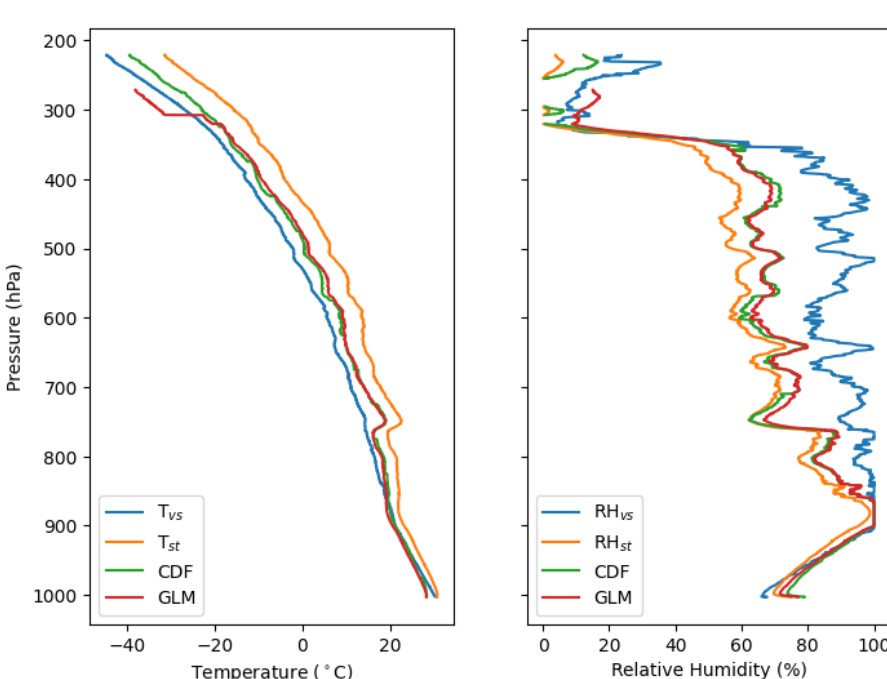


**Figure 11. The temperature (left) and RH (right) of the co-launch on 2018-08-27 06Z. The reference (VS) is illustrated in blue, the ST in range, CDF-corrected in green, and GLM-corrected in red.**

From the cases shown above, we also notice the characteristics of different correction methods. The CDF-corrected
temperatures show a wavy pattern along the pressure levels due to the bin-based correction. The GLM adjustments look like
horizontal shifts of the original values due to the linearity of the model.

Despite the simplicity of our correction methods, the temperature bias between ST and VS can be reduced from 3.0 K to 0.9
K and the RH bias from 8.5% to 6.9%. Note that our correction methods also reduce the standard deviations from 1.8 K to 0.6
K and 3.8% to 2.8%, respectively. Hence, we can expect 80% of ST observations to exhibit less than 1K bias in temperature
and 8.8% bias in RH.

The corrected ST measurements aligned well with the VS data, especially when the sounding successfully reached an altitude
higher than 300hPa. For those co-launches that ended early, though their bias is still low in statistics, their profiles usually
looked problematic when visualized. We recommend further looking into the reasons that cause the sounding to end early.



## 5.2 A use case of ST

The low cost of the ST can facilitate high spatial-temporal frequency of upper-air observations. While the ST provides reasonable measures after correction, its reliability in higher altitudes is still incompatible with the VS used in standard operation. Therefore, here, we demonstrate a use case to illustrate the strength of the ST. Figure 12 illustrates a set of continuous ST profiles on 2018-08-17 with one-hour intervals. Figure 12 shows the evolution of a local convective system, which is not feasible in regular 12-hour interval radiosonde operation -the increase of atmospheric moisture at 1300 local time before the

heavy rain occurrence is observed. Using the flexibility in deploying the ST during field campaigns allows us to capture vertical profiles in the troposphere at an hourly, or even shorter, time interval. This is notably advantageous for understanding the development of deep convection, which typically has a lifetime of 1 to 3 hours, and the surrounding environment, especially the lower boundary layer. A similar ST profile has been used in the study of the afternoon thunderstorm in Taipei compared to the results from CRESS cloud-resolving modeling (Tsujino et al. 2022). Note that the ST data here was corrected with the

CDF-based method; better performance can be achieved with GLM-based methods.

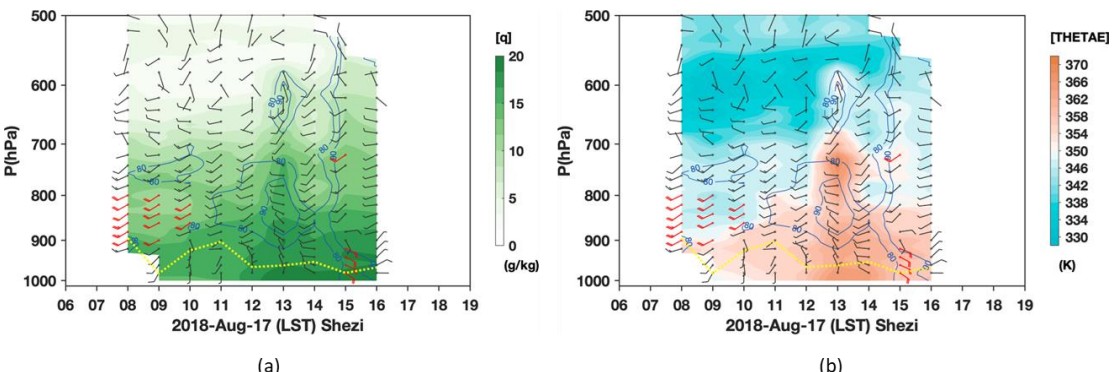

**Figure 12. The continuous ST observations of one-hour intervals on 2018-08-17 at Shezi. The soundings were corrected with CDF, and the derived specific humidity, q, is shown in panel (a) together with the wind field. The derived equivalent potential**

**temperature, Θe, is shown in panel (b).**

## 6. Concluding Remarks

In this study, we explored upper-air observations by assessing the capabilities and potential of the Storm Tracker (ST) as an alternative to traditional instruments like the Vaisala RS41-SGP (VS). The GPS estimated ST wind error difference is about

0.05 ms-1. To ensure the reliability of ST measurements in temperature and moisture, we conducted over a thousand co-launches of the ST and the VS, evaluating and refining the performance of the ST through developed correction methods for temperature and humidity measurements. The corrected ST soundings have 1-K temperature and 7% relative humidity root





mean square difference to the VS soundings. These error differences can be reduced to 0.66-K and 4.61% below the 700-hPa height.

Derived from the co-launch dataset, two correction methods based on CDF and GLM algorithms were implemented to enhance the quality of temperature and humidity observations in the ST. Both methods work comparably well to reduce the biases of the ST. While the CDF-based correction is robust and reliable, the GLMs easily model and change the predictors. And despite being lightweight and cost-effective, the ST exhibited observations closely aligned with the VS after corrections, particularly in the lower atmospheric layers below 500hPa. The geostrophic adjustment dynamics indicate that the spatial temperature

variation in the free atmosphere may not be large. However, the lower atmosphere may have large variations in temperature and moisture. This positions the ST as a promising candidate for supplementing regular upper-air observations for high spatial and temporal resolution in the lower atmosphere.

Although we used the linear regression version of GLMs in this study, the concept of modeling the joint probability distribution can be extended to various statistical models such as decision trees, support vector machines (SVM), and artificial neural

networks (ANN). The simple GLMs in this study assume the response is a Gaussian distribution of the linear combination of predictors. Other machine learning models can establish nonlinear mappings between the predictors and response without assuming any distributions. However, investigating more machine learning models is beyond the scope of this study.

In addition, while the VS remains the standard for upper-air observation, the cost-effectiveness and demonstrated efficacy of the ST post-correction mark a significant benefit for atmospheric research. The ST's adaptability and potential for integration

into scientific field campaigns or standard operational practices showcase its value. As hardware enhancements and more sophisticated correction methodologies for the ST are anticipated, its capacity to contribute significantly to atmospheric studies is poised for growth.

In closing, the ST represents a beacon of innovation in observational technology and reflects the evolving landscape of meteorological research. Its adaptability, affordability, and close approximation to the VS make it a suitable alternative for

high spatial and temporal profiles of lower atmospheric observations. Ongoing advancements in hardware and correction methods solidify the ST's role as an asset for scientific field observations.

**Code Availability**

Code for data cleaning and analysis is provided as part of the replication package. It is available at https://www.dropbox.com/scl/fo/ah7i6z4f7u2yzijfh7ua3/h?rlkey=ar4g2hq7hwkop2eyzw83el8ih&dl=0 for review. It will be

uploaded to GitHub once the paper has been conditionally accepted.

**Data Availability**

The data for this project are confidential but may be obtained with Data Use Agreements with the National Taiwan University. Researchers interested in access to the data may contact Ting-Shuo Yo at tsyo@ntu.edu.tw. It can take some months to



negotiate data use agreements and gain access to the data. The author will assist with any reasonable replication attempts for
two years following publication.

**Author Contribution**

Kuo, Yo, and Yu contributed equally to this article's composition, while Su provided thorough reviews of prior studies. Kuo, Yu, Su, Liu, and Lin coordinated and conducted the field observations throughout the seven years and collected the invaluable co-launch dataset. Yo and Yu are significant contributors to the presented data preprocessing and correction methods, and Yo
is responsible for the design and execution of the analysis.

**Competing interests**

The authors declare that they have no conflict of interest.

**Acknowledgment**

This study was supported by the National Science and Technology Council (NSTC) of Taiwan for many years of support under
Grants MOST 107-2628-M-002-016, MOST 108-2119-M-002-022, MOST 109-2111-M-002-008, MOST 110-2123-M-002-007, NSTC 111-2123-M-002-014, and NSTC 112-2123-M-002-006.

We want to express our sincere gratitude to Wei-Chun Huang, who contributed to developing the storm tracker in the first place. Our heartful appreciation to the Central Weather Administration (CWA) for most of the co-launch in Vaisala RS41-SGP (VS) and Storm Tracker (ST). We thank the research team members of the Taipei Summer Storm Experiment (TASSE),
the Yilan Experiment of Severe Rainfall (YESR2020), the Taiwan-Area Heavy Rain Observation and Prediction Experiment (TAHOPE), the Northern Coast Observation, Verification, and Investigation of Dynamics (NoCOVID21), and the Mountain Cloud Climatology (MCC) project. Their dedication and commitment were instrumental in the realization of our research objectives.

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
