# Peer review of "Data quality control and calibration for mini-radiosonde system "Storm Tracker" in Taiwan"

_EGUsphere, 2024_

## Referee Comment (RC1)

The manuscript *"Data quality control and calibration for mini-radiosonde system Storm Tracker in Taiwan"* describes a twin sounding campaign of the Storm Tracker and the Vaisala RS41 radiosonde, where the latter is used as reference. The results of the twinsoundings are used to derive a correction for the Storm Tracker temperature and humidity profiles, using a statistical method based on the cumulative distribution function (CDF). I have considerable concerns with the way the twinsoundings were performed (payload configuration) as well as the applied analysis method, both of which I think affect the results and conclusions of this study.

One concern is with the method applied to analyse the data from the coincident twinsoudings. Since these soundings are performed with two different radiosondes on the same balloon, this allows for a direct comparison of the profile data. Using a statistical method like the cumulative distribution function (CDF) seems to me like an unnecessary complication. To my understanding the CDF-based method as employed by Ciesielski et al., is applied when comparing radiosonde data taken under similar meteorological conditions albeit not coincidently on the same rig. The advantage of coincident twinsounding data is that it allows to directly determine the bias + associated uncertainty between both systems. Furthermore, the physical mechanism that is causing the bias between both radiosondes is warming by solar radiation that is counteracted by convective cooling by the air flowing over the sensor (ventilation). The efficiency of the convective cooling is directly linked to the altitude-dependent ambient air pressure. The CDF method that is applied in the manuscript does indeed derive corrections for pressure ranges, but the purpose of further analysing the differences in sense of ambient temperature is not clear to me.

For a description of the Storm Tracker radiosonde and its sensors, the reader is referred to another publication (Hwang et al. 2020) . However, for a good understanding and interpretation of the results presented in this manuscript a (brief) description of the radiosonde design, and specifications of the sensor are essential. Also a description of the payload configuration for the twinsoundings is necessary, currently the reader has to deduce the payload configuration from a rather poor-quality (underexposed) photograph. From the photograph in Figure 1 it becomes clear that the two radiosondes are connected back-to-back. The sensor boom of the RS41 is pointing upwards with the T + RH sensors probing undisturbed and uncontaminated air. However the integrated T & RH sensor of the Storm Tracker radiosonde is located about 20cm lower, directly in front of the radiosonde's housing. As a result the air flowing over the T & RH sensors is very likely contaminated by the housing/casing of the radiosonde, affecting the measurements. This payload configuration most likely explains the large temperature between the Storm Tracker and the RS41 shown in Figure 2. I was very surprised by the large temperature difference between both radiosondes (approx. 5 K at ground level), and my first thought was that this was caused by a calibration error of the temperature, but this bias is not present for the nighttime flight presented in Figure 10, so that it is indeed likely that this large bias results from solar radiation.

It can't be excluded that the large differences between Storm Tracker and RS41 are caused by the payload configuration, rather than by the performance of the individual sonde types. This raises the question whether the observed differences in this comparison study reflect differences that would be observed between both radiosondes when flying on separate balloons, or on payloads that are better configured for comparisons. Therefore, I am not sure whether the results found in this study are representative for the differences/bias between both radiosonde types and to my opinion it is doubtful that this study can be used to derive a generic correction for T & RH profiles of the Storm Tracker radiosonde.

If the temperature error due to solar radiation is that large, 5 K at ground level and up to 10 K at higher altitudes, it also inevitably limits the quality of the temperature profile after correction, i.e. there will be a considerable associated uncertainty remaining. In such a situation the preferred strategy is to improve the design of the temperature sensor so that it is less sensitive to solar radiation.

While reading the manuscript I noticed various other things that I will mention below and that the authors may consider well-meant advice for a future publication. Since the study described in this manuscript has some fundamental flaws, I consider it unlikely that it will go to the stage of e.g. major revisions.

- The literature references provided in the introduction all are quite old, and recent work on the characterisation of solar radiation temperature error is missing. Important progress in this field is made by the efforts of the GRUAN community, which should be mentioned in the introduction. Suggested publications are for example:

- – von Rohden et al AMT2022 (doi 10.5194/amt-15-383-2022)

- – Lee et al. AMT2022 ( doi 10.5194/amt-15-1107-2022)

- – Hoshino et al. AMT2022 (doi 10.5194/amt-15-5917-2022)

- – GRUAN-TD-5 (https://www.gruan.org/documentation/gruan/td/gruan-td-5)

- Briefly describe the characteristics of the radiosonde and its sensors. A more detailed description can indeed be provided in another paper.

- Use a regular plot for the comparison of radiosonde profiles, instead of a Skew-T diagram such as in Figure 2.

- In addition to Table 1, show a map with the location of the sites.

- Table 2 appears very full. Leave out the zeroes, since they distract and don't provide any useful information. Alternatively use another way of presenting these data, such as a e.g. a bar chart.

- Figure 8: separate for daytime and nighttime data.

- The profile shown in Figure 11 exhibits an interesting discrepancy between RS41 and Storm Tracker around 800 hPa. There is a wiggle in the Storm Tracker's temperature profile that is not seen for the RS41. It coincides with a decrease in RH. Is this a radiation effect connected to a cloud top?

---

## Author Comment (AC1)

**Replies to reviewers' comments**

**Data quality control and calibration for mini-radiosonde system "Storm Tracker" in Taiwan (egusphere-2024-661)**

We really appreciate the comments to the manuscript from the two reviewers. We have carefully reviewed and adjusted accordingly, and prepared the replies to each comment as follows (the original comments are in blue, replies in black).

**Reviewer 1:**

… One concern is with the method applied to analyse the data from the coincident twinsoudings. Since these soundings are performed with two different radiosondes on the same balloon, this allows for a direct comparison of the profile data. Using a statistical method like the cumulative distribution function (CDF) seems to me like an unnecessary complication. To my understanding the CDF-based method as employed by Ciesielski et al., is applied when comparing radiosonde data taken under similar meteorological conditions albeit not coincidently on the same rig. The advantage of coincident twinsounding data is that it allows to directly determine the bias + associated uncertainty between both systems. Furthermore, the physical mechanism that is causing the bias between both radiosondes is warming by solar radiation that is counteracted by convective cooling by the air flowing over the sensor (ventilation). The efficiency of the convective cooling is directly linked to the altitude-dependent ambient air pressure. The CDF method that is applied in the manuscript does indeed derive corrections for pressure ranges, but the purpose of further analysing the differences in sense of ambient temperature is not clear to me.

For a description of the Storm Tracker radiosonde and its sensors, the reader is referred to another publication (Hwang et al. 2020) . However, for a good understanding and interpretation of the results presented in this manuscript a (brief) description of the radiosonde design, and specifications of the sensor are essential. Also a description of the payload configuration for the twinsoundings is necessary, currently the reader has to deduce the payload configuration from a rather poor-quality (underexposed) photograph. From the photograph in Figure 1 it becomes clear that the two radiosondes are connected back-to-back. The sensor boom of the RS41 is pointing upwards with the T + RH sensors probing undisturbed and uncontaminated air. However the integrated T & RH sensor of the Storm Tracker radiosonde is located about 20cm lower, directly in front of the radiosonde's housing. As a result the air flowing over the T & RH sensors is very likely contaminated by the

housing/casing of the radiosonde, affecting the measurements. This payload configuration most likely explains the large temperature between the Storm Tracker and the RS41 shown in Figure 2. I was very surprised by the large temperature difference between both radiosondes (approx. 5 K at ground level), and my first thought was that this was caused by a calibration error of the temperature, but this bias is not present for the nighttime flight presented in Figure 10, so that it is indeed likely that this large bias results from solar radiation.

It can't be excluded that the large differences between Storm Tracker and RS41 are caused by the payload configuration, rather than by the performance of the individual sonde types. This raises the question whether the observed differences in this comparison study reflect differences that would be observed between both radiosondes when flying on separate balloons, or on payloads that are better configured for comparisons. Therefore, I am not sure whether the results found in this study are representative for the differences/bias between both radiosonde types and to my opinion it is doubtful that this study can be used to derive a generic correction for T & RH profiles of the Storm Tracker radiosonde.

If the temperature error due to solar radiation is that large, 5 K at ground level and up to 10 K at higher altitudes, it also inevitably limits the quality of the temperature profile after correction, i.e. there will be a considerable associated uncertainty remaining. In such a situation the preferred strategy is to improve the design of the temperature sensor so that it is less sensitive to solar radiation.
* * *
We acknowledge that the design of our co-launch experiment differs from some previous field campaigns. In this study, attaching the Storm Tracker (ST) to the Vaisala radiosonde (VS) allowed for direct inter-comparison at native temporal resolution (every second). Additionally, the Cumulative Distribution Function (CDF) is a well-established non-parametric statistical model for regression and classification within the community. The concept of correction processes for temperature (based on different pressure levels) and relative humidity (based on different temperature levels) between different instruments is similar to methods used in other field campaigns (e.g., Ciesielski et al. 2014). Furthermore, we compared CDF with a parametric alternative, Generalized Linear Models (GLM), for data correction purposes in this study. Our results indicate that both models perform well for this task.

In over 1,000 co-launches, we consistently bound ST and VS in the same configuration, as we believe this is essential for a controlled experiment. While it is challenging to definitively prove that the observed biases are unrelated to the binding method, we have conducted thorough checks, including in-lab measurements of both sensors and several co-launches with the sensors in separate balloons (although this data was not used for correction). We found that the bias patterns observed were consistent with those in the co-launch dataset.

Regarding the larger temperature biases near ground level, these issues are primarily due to the sun directly heating the ST while waiting to launch. In most cases, we prevent this, and the correction table excludes such instances as outliers.

We must also acknowledge that we cannot entirely rule out the possibility that the payload configuration (especially the casing) could contaminate the T and RH measurements of ST. The overall idea behind the ST hardware design is to leverage easily-accessible commercial sensors and serve as a supplement to regular sounding observations, specifically in the lower boundary layer. While further hardware upgrades are underway, the ST cannot replace the more mature design instruments. Therefore, we suggest co-launches for future applications to

ensure data accuracy.

*Ciesielski, P. E., Yu, H., Johnson, R. H., Yoneyama, K., Katsumata, M., Long, C. N., ... & Van Hove, T. (2014). Quality-controlled upper-air sounding dataset for DYNAMO/CINDY/AMIE: Development and corrections. Journal of Atmospheric and Oceanic Technology, 31(4), 741-764.

The literature references provided in the introduction all are quite old, and recent work on the characterisation of solar radiation temperature error is missing. Important progress in this field is made by the efforts of the GRUAN community, which should be mentioned in the introduction.
Suggested publications are for example:
– von Rohden et al AMT2022 (doi 10.5194/amt-15-383-2022)
– Lee et al. AMT2022 ( doi 10.5194/amt-15-1107-2022)
– Hoshino et al. AMT2022 (doi 10.5194/amt-15-5917-2022)
– GRUAN-TD-5 (https://www.gruan.org/documentation/gruan/td/gruan-td-5)

We have reviewed the suggested references and added them to the article.

Briefly describe the characteristics of the radiosonde and its sensors. A more detailed description can indeed be provided in another paper.

The following paragraph was added to section 1 for more details on the ST sensors. Again, specific hardware details are described Hwang et al. 2020.

"The ST consists of a microcontroller (ATMEGA328p), a GPS sensor (U-blox MAX7-Q), a pressure sensor (Bosch BMP280), a temperature–humidity sensor (TE-Connectivity HTU21D), and a transmitter (LoRa™). The sensors have an overall operation range from 1100 to 300 hPa in pressure and from -40℃ to 85℃ in temperature. The ST used a regular AAA battery for 2-4 hours of power; the total weight was 20g. More detailed hardware specifications can be found in Hwang et al., 2020. The design of ST aimed to leverage the low cost of sensors used in commercial electronics to enable high-frequency observations in the boundary layer. The receiver was designed to receive up to six STs simultaneously. It was ideal to use ST to gather supplemental data between regular sounding."

Use a regular plot for the comparison of radiosonde profiles, instead of a Skew-T diagram such as in Figure 2.

The new figure 2 as follows is updated as suggested.

[Figure]

2018-06-26 03Z(UTC)

In addition to Table 1, show a map with the location of the sites.

New Figure 3 is added as requested.

[Figure]

Table 2 appears very full. Leave out the zeroes, since they distract and don't provide any useful information. Alternatively use another way of presenting these data, such as a e.g. a bar chart.

The bar chart as shown below is too dense for the information. Instead, we took the suggestion and left out the zeroes and updated the table as follows.

[Figure]

| Month | 2018 Day | 2018 Night | 2019 Day | 2019 Night | 2020 Day | 2020 Night | 2021 Day | 2021 Night | 2022 Day | 2022 Night | Total Day | Total Night |
|---|---|---|---|---|---|---|---|---|---|---|---|---|
| 1 | | | | | 21 | 24 | | | 25 | 26 | 46 | 50 |
| 2 | | | | | 29 | 29 | | | | | 29 | 29 |
| 3 | | | | | 27 | 31 | | | | | 27 | 31 |
| 4 | | | | | 30 | 30 | 15 | 13 | | | 45 | 43 |
| 5 | | | | | 6 | 5 | 6 | 4 | | | 12 | 9 |
| 6 | 14 | | 20 | | | | 30 | 32 | | | 64 | 32 |
| 7 | 14 | | 60 | 12 | | | 22 | 23 | | | 96 | 35 |
| 8 | 41 | | 85 | | | | 23 | 22 | | | 149 | 22 |
| 9 | | | | | | | 25 | 26 | | | 25 | 26 |
| 10 | | | | | | | 29 | 28 | 7 | 3 | 36 | 31 |
| 11 | | | | | 20 | 17 | 40 | 41 | 6 | 7 | 66 | 65 |
| 12 | | | | | | | 30 | 31 | | | 30 | 31 |
| Total | 69 | 0 | 165 | 12 | 133 | 136 | 220 | 220 | 38 | 36 | 625 | 404 |

Figure 8: separate for daytime and nighttime data.

Day- and night-time analyses are separated as suggested.

[Figure]

[Figure]

The profile shown in Figure 11 exhibits an interesting discrepancy between RS41 and Storm Tracker around 800 hPa. There is a wiggle in the Storm Tracker's temperature profile that is not seen for the RS41. It coincides with a decrease in RH. Is this a radiation effect connected to a cloud top?

Though we cannot rule out the possibility of radiation effect, we tend to believe this belongs to a series of special cases when conducting co-launch during extreme weather. As described in section 5.1, a severe rainfall event was caused by the convergence of the tropical depression and the southwest monsoon from August 23 to August 30, 2018. The bias between ST and VS during this event was larger than usual. We presented this case to demonstrate the "worst case" scenario for using ST (well, it is not literally the worst case, but the 6th worst one if we didn't remove co-launches with too few records).

---

## Author Comment (AC2)

**Replies to reviewers' comments**

**Data quality control and calibration for mini-radiosonde system "Storm Tracker" in Taiwan (egusphere-2024-661)**

We really appreciate the comments to the manuscript from the two reviewers. We have carefully reviewed and adjusted accordingly, and prepared the replies to each comment as follows (the original comments are in blue, replies in black).

**Reviewer 2:**

… While making more than 1000 dual flights with VS to deduce correction tables/functions for the ST measurements is a very good effort and the results are worth being reported in AMT, I tend to feel that the current manuscript is "overselling" the ST. This is mainly because the temperature and RH sensors of the ST, as shown in Figures 2 and 3 of Hwang et al. (2020), are never optimal for balloon upper-air sounding, e.g. with possible large contaminations to the temperature measurements depending on solar radiative heating and thus cloud cover status as well as day-versus-night difference. I would also have concern about the production stability of the sensor, i.e. whether the characteristics and the quality are within the reasonable range of uncertainty in different production batches (e.g. in different years). This may mean that in the end, we may always need dual flights with a well characterized and reliable radiosonde like the VS. In addition, the radiowave used for ST is from 432 to 436.5 MHz (Hwang et al., 2020) which is for amateur radiolocation, not for meteorological aids (e.g. International Telecommunication Union (ITU), "Radio Regulations", Volume 1, 2020, available from https://www.itu.int/pub/R-REG-RR-2020); thus, in some countries ST cannot be used as a meteorological radiosonde officially. This may mean that the ST cannot be "widely" used in the future. Note also that one-hourly sounding campaigns like the one shown in Section 5.2 are not impossible with modern radiosondes like the VS; thus it is not easy for me to imagine possible applications of the ST aiming at new scientific studies that are only possible with the ST.

    Regarding the issue of overselling, it is important to clarify that while Soundings from Triggers (ST) are indeed suitable for Planetary Boundary Layer (PBL) studies and areas with complex terrain, they should be viewed as a supplement to Vaisala Soundings (VS) rather than a replacement. Atmospheric conditions at higher altitudes may not always require high-frequency observations due to geostrophic adjustment, but ST can significantly enhance our

understanding of PBL dynamics.

In regions such as Southeast Asia, where PBL conditions can vary within short distances, the high spatial frequency observations provided by ST are particularly beneficial. It is important to note, however, that the correction results presented are specific to Taiwan. To ensure broader applicability, we suggest conducting co-launches during field campaigns. This approach would allow users to derive in-situ correction formulas using the proposed methods.

Furthermore, we recommend utilizing ST between VS launches to optimize data collection and analysis. This combined approach will enhance the overall effectiveness of atmospheric observations and improve the accuracy of data interpretation. New statements are added in section 6 to address this.

Finally, regarding radio regulations, the hardware design for STs allows for adjustments to different radio bands if needed. While a hardware update is beyond the scope of this manuscript, it is important to note that radio regulation should not pose an issue for field campaigns using STs.
* * *
Line 16 (and line 369): I am afraid that "geostrophic adjustment dynamics" is never discussed in the manuscript.
* * *
We added a few sentences to elaborate our point.

"For synoptic weather, geostrophic adjustment dynamics suggest that spatial temperature variations in the free atmosphere may not be significant, reducing the need for high-frequency upper-air radiosonde observations. Consequently, most operational radiosonde observations worldwide are conducted daily at 00Z and 12Z, with intervals of 12–24 hours. However, atmospheric phenomena originating from the boundary layer are often smaller in scale and closely related to local terrain. For example, a single convective cell typically lasts for minutes, while thunderstorms persist for a few hours. To gain a better understanding of these types of weather, a low-cost device capable of deploying multiple sensors simultaneously or at intervals of less than an hour can enhance field experiments. This approach provides valuable insights into the lower atmosphere's significant variations in temperature and moisture, especially for convective systems that may lead to disastrous rainfall or flash flooding."
* * *
Introduction and Figure 1: Please describe the main technological points of the ST in more detail, including the model and characteristics of the equipped temperature-RH sensor. Figure 1(b) should be much greater, and an enlarged photo for the sensor part may be added. Also, Figure 1(c) is not a good one, because the ST sensor boom (or "sensor box") is not well shown.
* * *
We added more detailed information about the ST in the introduction section. We didn't modify Figure 1 as suggested because the Hwang et al., 2020 paper had the same figure, and we want to avoid copyright issue.

We added:
"The ST consists of a microcontroller (ATMEGA328p), a GPS sensor (U-blox MAX7-Q), a pressure sensor (Bosch BMP280), a temperature–humidity sensor (TE-Connectivity HTU21D), and a transmitter (LoRa™)."

Introduction, the review part of various radiosonde issues: The papers cited here tend to be too old. More recent papers for more recent radiosondes need to be cited. These include:
– Vaisala RS41 radiosonde (but with GRUAN data processing): Sommer et al., GRUAN characterisation and data processing of the Vaisala RS41 radiosonde. GRUAN Technical Document 8 (GRUAN-TD-8), v1.0.0 (2023-06-28), https://www.gruan.org/documentation/gruan/td/gruan-td-8.
– Modem M10 radiosonde: Dupont, J., M. Haeffelin, J. Badosa, G. Clain, C. Raux, and D. Vignelles, 2020: Characterization and Corrections of Relative Humidity Measurement from Meteomodem M10 Radiosondes at Midlatitude Stations. J. Atmos. Oceanic Technol., 37, 857–871, https://doi.org/10.1175/JTECH-D-18-0205.1.
– Meisei RS-11G and iMS-100 radiosondes: Kizu et al., Technical characteristics and GRUAN data processing for the Meisei RS-11G and iMS-100 radiosondes, GRUAN-TD-5, v1.0 (2018-02-21), https://www.gruan.org/documentation/gruan/td/gruan-td-5.
These also include very useful information on the modern radiosonde sensor characteristics and on all necessary corrections to the radiosonde measurements.

We have reviewed the suggested references and added them to the introduction.

Figure 2: Are the ST RH measurements really dry biased in comparison with the VS measurements below the 500 hPa level? They look wet biased in this case.

We followed the suggestions from other reviewers to replace the skew-T-log-p plot with a regular plot. The aspect ratio of our previous plot could be hard to read and misleading.

As illustrated in the new figure, the ST showed a dry bias in general except some mixed bias pattern above 800hPa.

[Figure]

2018-06-26 03Z(UTC)

Lines 98-105: Testing mathematically more sophisticated machine-learning-based methods is good and interesting, but if the sensor characteristics are not optimal for upper air sounding (e.g. ~5 deg.C temperature error as shown in Figure 2 is just too large!), I tend to think that we should improve the sensor itself before reconsidering correction methods.

We totally agree. Hence, a newly designed ST with different sensors is currently undergoing. However, the improvement of the hardware design is not within the scope of the presented work.

Table 1: Please also add the information on e.g. climate zone, season, etc., i.e. the information listed at Lines 92-93, to the place e.g. right after Location.

We added a map with co-launch sites marked on it. All sites are in the same climate zone. As shown on the map, those sites are of similar altitudes but with different terrain.

[Figure]

Figure 3: I confused with the three horizontal lines within the gray box. I thought that the data processing of the ST and the VS is independent before "L2_ST-VS". In other words, I thought that the "Paired Entries" are used to establish correction tables/functions with various different methods. Or, in other words, I thought that for the VS, the authors simply use the manufacturer-processed data set, while I have impression from the current figure that the authors make their own (and perhaps common) corrections from L1_VS to L2_VS. The term Level 2 may be confusing if it is used for the ST at this stage, because the ST data will be

further corrected by using the VS data as the reference if my understanding is correct.

We have modified the figure to avoid confusion. The correction from level 1 to level 2 is done separately for ST and VS. These procedures aim to remove missing values and corrupted data. While VS data is exactly the manufacturer-processed data set, the procedure of L1_ST to L2_ST is done by implementing the specifications described in Hwang et al., 2020.

We want to position our work as a step between level 2 and level 3 data, in which we "identify biases in data and correct if possible" (Ciesielski et al., 2011) by aligning to the VS data using statistical methods.

[Figure]

Section 3.1 (and Figure 4 and its caption): Please add the explanation how to obtain delta T (and its uncertainty) from the CDF.

We added:

"The observed temperature records are sorted in ascending order, and then the proportion of observations is derived for every 0.1-degree interval from -80 to 40 degrees Celsius as the probability density."

Line228: three-dimensional?

Yes, if we define the correction of RH as a function of temperature and pressure, the joint

probability distribution of p(P, T, RH) is three-dimensional.

Lines 241-244: Please also add "(GLM1)", "(GLM2)", and "(GLM3)" here, not later.

We added the notes as suggested.

Lines 255-258: Solar elevation angle may be a better variable?

Yes, it is. And we even calculated the direct solar radiation as a predicting variable. However, we wanted to limit the scope of the presented work to the feasibility and recommended procedure of using ST for high-frequency boundary layer observations. Hence, the correction methods presented involved only basic statistics and data that are directly accessible.

We had another ongoing project for finding the best correction methods including both statistical models and predicting variables.

Figure 7: Results from which method, CDF, GLM*?

Figure 7 demonstrated the bias between VS and ST before correction. We modified the caption to avoid confusion.

Lines 283-285: I have a difficulty on this discussion. I am not an expert of metrology (i.e. science of measurement), but I think that that the "average errors" here are comparable to the uncertainties of the VS measurements does not mean that the uncertainties of the ST measurements (after the corrections) are comparable to the uncertainties of the VS measurements. Can we say roughly how large are the uncertainties of the corrected ST measurements in this case? Is it ~1.414 times?

The "average error" (we modified it to "averaged RMSE") means the bias of ST after correction. As for the uncertainties of "ST after correction" itself, it should be 1:1 in magnitude to the term "stdev" (in the right side of Table 3), which represents the "range of variation of the variable".

The "errors" reported here are Root-Mean-Squared-Error (RMSE, the square root of the mean squared error), which is a common measurement for evaluating predicted values. The "squared" and "root" are used to ensure the positive and negative deviations will not cancel out each other, and hence this value should be 1:1 in magnitude to uncertainties.

Figures 10 and 11: The authors should also show delta T and delta RH profiles as well. In particular, delta T profiles are very important because we need tropospheric temperature measurements with uncertainties less than e.g. 0.5 K or even 0.1 K usually, and this degree of differences is hard to see with the current panels.

We modified the figure as suggested. The bias in temperature of corrected ST can be as low as 0.66 degree below 700hPa in average with stdev of 0.34. This means the 95% of corrected T having bias between 0 to 1 degree.

[Figure]

Line 330: A wavy pattern. Which pattern are the authors referring to? With this figure, I am afraid that we cannot judge that GLM may be better.

We were not arguing the GLM is "better" here. We simply pointed out the characteristics of different statistical models. While the CDF approach used separate bins, the corrected profile will showed segments corresponding to the bins. We admitted such segments were hardly noticeable if one doesn't bear this concept in mind, and hence we removed the description to avoid confusion.

Section 5.2: One-hourly sounding campaign is possible with the VS as well.

Yes, it is possible, but with a much higher price. We didn't argue that ST can do things that VS cannot do, but ST can provide data of reasonably good quality with a much lower cost. Furthermore, STs are also capable of simultaneous observations with just one system, which enhance the flexibility of field campaigns.

Line 369: Geostrophic adjustment dynamics has never been discussed, I am afraid.

We added a few sentences to elaborate our point.

"For synoptic weather, geostrophic adjustment dynamics suggest that spatial temperature variations in the free atmosphere may not be significant, reducing the need for high-frequency upper-air radiosonde observations. Consequently, most operational radiosonde observations worldwide are conducted daily at 00Z and 12Z, with intervals of 12–24 hours. However, atmospheric phenomena originating from the boundary layer are often smaller in scale and closely related to local terrain. For example, a single convective cell typically lasts for minutes, while thunderstorms persist for a few hours. To gain a better understanding of these types of weather, a low-cost device capable of deploying multiple sensors simultaneously or at intervals of less than an hour can enhance field experiments. This approach provides valuable insights into the lower atmosphere's significant variations in temperature and moisture, especially for convective systems that may lead to disastrous rainfall or flash flooding."

Lines 373-377: As discussed above, I personally think that the sensor characteristics (including the sensor covering and orientation) need to be improved and more optimized for upper-air sounding, before considering mathematically more sophisticated correction methods.

Yes, we agree. And as mentioned in the earlier response, the next generation of ST hardware design is under development. The represented work focused on the evaluation and recommended procedure of the current ST hardware.

Lines 378-386: As discussed in the beginning, I tend to think that the authors are overselling the ST here.

We modified the corresponding paragraphs to avoid the overselling. The full response to this point is in the beginning.